# Multi-Object Multi-Camera Tracking Based on Deep Learning for Intelligent Transportation: A Review

**DOI:** 10.3390/s23083852

**Published:** 2023-04-10

**Authors:** Lunlin Fei, Bing Han

**Affiliations:** 1School of Traffic and Transportation, Beijing Jiaotong University, Beijing 100044, China; 2Jiangxi Provincial Transportation Investment Group Co., Ltd., Nanchang 330029, China; 3School of Civil Engineering, Beijing Jiaotong University, Beijing 100044, China; bhan@bjtu.edu.cn

**Keywords:** multi-object multi-camera tracking, deep neural network, object detector, intelligent transportation

## Abstract

Multi-Objective Multi-Camera Tracking (MOMCT) is aimed at locating and identifying multiple objects from video captured by multiple cameras. With the advancement of technology in recent years, it has received a lot of attention from researchers in applications such as intelligent transportation, public safety and self-driving driving technology. As a result, a large number of excellent research results have emerged in the field of MOMCT. To facilitate the rapid development of intelligent transportation, researchers need to keep abreast of the latest research and current challenges in related field. Therefore, this paper provide a comprehensive review of multi-object multi-camera tracking based on deep learning for intelligent transportation. Specifically, we first introduce the main object detectors for MOMCT in detail. Secondly, we give an in-depth analysis of deep learning based MOMCT and evaluate advanced methods through visualisation. Thirdly, we summarize the popular benchmark data sets and metrics to provide quantitative and comprehensive comparisons. Finally, we point out the challenges faced by MOMCT in intelligent transportation and present practical suggestions for the future direction.

## 1. Introduction

MOMCT is a crucial problem in computer vision, and it is very useful in public safety. MOMCT aims to track multiple vehicles or other objects in traffic scenes through multiple cameras, which is different from MOT (MOT) in a single camera [1]. A camera network consisting of several cameras has a wider field of view than a single camera and offers more practical application prospects. The main application scenarios include vehicle cross-regional tracking on smart highway, traffic management in smart cities [2], autonomous driving [3], and crowd analysis [4]. Especially in the process of vehicle cross-regional tracking on the highway, the MOMCT task can be used to track multiple vehicles simultaneously, which plays a key role in traffic management and analysis. Therefore, most algorithms used in MOMCT are based on the MOT algorithm, such as feature extraction algorithm, object modeling, and motion detection. The MOMCT system consists of two components: firstly, all the objects in each video frame are tracked and located by a single camera, and the detection output results are connected into a continuous trajectory across time; secondly, the network composed of different cameras matches the accurate vehicle trajectory detected across different cameras through the correlation module. The vehicle cross-regional tracking process of MOMCT system is shown in Figure 1.

In the real scenes, besides background confusion [5], posture change [6] and occlusion [7], there are still many difficulties in tracking and detecting objects because of the different video quality, lighting and viewing angle of each camera. In addition, multiple cameras will not share the overlapping area. It means that the appearance features of the same object will be very different in cameras with different perspectives. In the stage of vehicle tracking, there are often problems such as large intra-class variability and inter-class similarity [8]. To solve these problems, most MOMCT methods follow the detection and tracking paradigm: firstly, a set of detection is generated independently for each video frame shot by the camera; secondly, these detections are linked together by similarity measurement to generate a continuous trajectory. Usually, this similarity measure considers the location information and visual characteristics of the object. Visual features are very important for keeping the side of the tracking object.

The vehicle tracking problem in multi-camera system is an extension of the problem in single-camera system. Therefore, the algorithms used in multi-camera tracking are mostly based on the well-known algorithms in single-camera tracking, such as motion detection [9], object modeling [10], and feature extraction [11]. Using multiple cameras has many advantages over using a single camera. It can mainly reduce errors caused by occlusion or other sensors. However, vehicle tracking in multi-camera systems is very challenging, because the tracking process must ensure the integration of information from different sensors. The system designed to deal with MOMCT tasks usually consists of five sub-modules, namely, object detection [12], multi-object single-camera tracking [13], vehicle re-identification (re-ID) [14], and multi-object multi-camera tracking [15,16]. The general process can be summarized as follows: Firstly, the vehicle detection module outputs vehicle coordinates and categories in units of frames. Then, based on the vehicle position and the learned features, the single-camera tracking and detection module generates candidate trajectories for every single camera. Finally, these candidate trajectories are matched on different cameras by associating the object with the global identity.

In the existing reviews [17,18,19,20], their work is more focused on reviewing multi-object multi-camera tracking methods using a coarse classification, involving a wide range of application scenarios lacking relevance. In addition, the current reviews lack references to advanced results from the last three years, thus failing to provide a comprehensive overview of the latest developments in MOMCT. Therefore, as shown in Figure 2, this paper provides an overview of recent advances in MOMCT-related technologies from different aspects for intelligent transportation applications, including four major issues such as object detection, object tracking, vehicle re-identification, and multi-target cross-camera. In addition, this paper further details their technical challenges and compares different solutions. Last but not least, we examine the performance of relevant MOMCT algorithms on various datasets by focusing on data comparisons and explore the potential value of these methods in smart cities. The main contributions of this paper are as follows:We provide a comprehensive overview of the application based on deep learning technology in multi-object multi-camera tracking tasks. We have classified and summarised the different stages of deep learning-based MOMCT algorithms, including object detection, object tracking, vehicle re-identification and multi-object cross-camera tracking.We have aggregated the most commonly benchmark datasets and standard metrics for MOMCT. We have combined various data for experimental visualisation and comprehensive metrics evaluation of the main algorithms in MOMCT.We discuss the challenges MOMCT has faced in recent years from several perspectives, as well as the main application scenarios in practice, and explore potential future directions for MOMCT.

**Figure 2 sensors-23-03852-f002:**
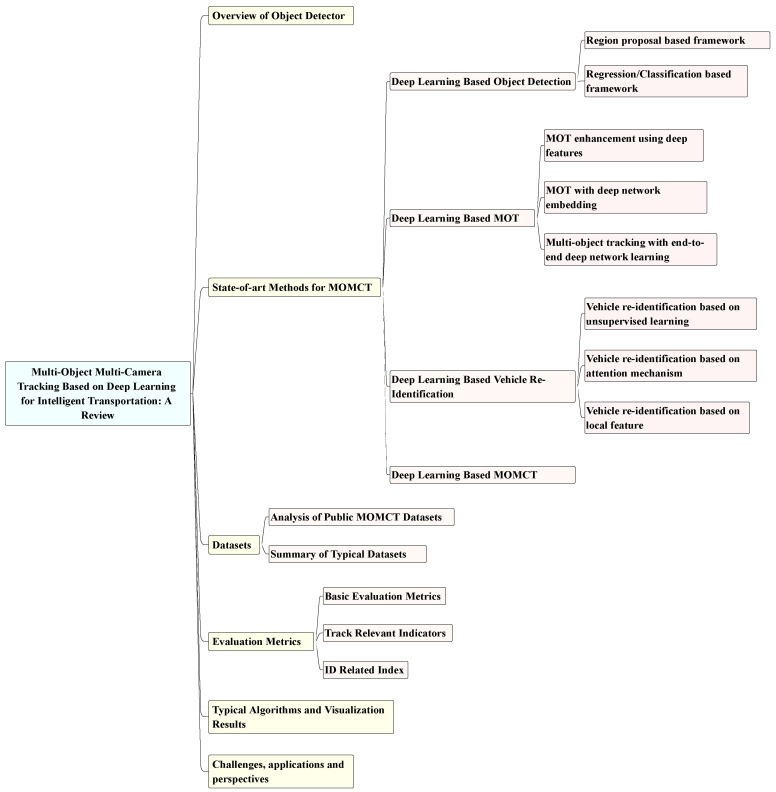
Hierarchical structure of the MOMCT based on deep learning.

Section 2 presents a typical baseline of MOMCT; Section 3 describes the classification of MOMCT-related algorithms based on deep learning; Section 4 describes the main datasets based on the MOMCT task; Section 5 describes in detail the evaluation criteria for MOMCT tasks; Section 6 shows typical algorithms and visualisation results for MOMCT; Section 7 details future challenges, practical application scenarios and future directions for MOMCT tasks; Finally, Section 8 concludes our work.

## 2. Overview of Object Detector

Object detection consists of two sub-tasks: localisation and classification. Localisation aims to determine the position of the object object in the video or image, while classification refers to the assignment of categories to the detected object objects (e.g., “vehicle”, “pedestrian”, “house ”, etc.). Figure 3 illustrates a detailed classification of deep learning-based object detectors. The classification and role of these object detectors will be discussed in this section.

### 2.1. Two-Stage vs. Single-Stage Object Detectors

In the field of object detection, the secondary object detector, such as R-CNN [21], Fast R-CNN [22] and Faster R-CNN [23], consists of two processes: candidate regions and object classification. In the stage of candidate region, the object detector selects regions of interest (ROIs) in input image contained object objects. In the stage of object classification, the most possible ROI is selected, other ROI is discarded, and the selected object is classified [24]. In contrast to two-stage detectors, single-stage object detectors create bounding boxes during object detection and perform classification operations on the detected objects. The advantage of single-stage detectors is that they are faster than two-stage detectors, but the disadvantage is that they are less accurate in comparison. Popular single-stage detectors include YOLO, SSD, and RetinaNet.

Figure 4 illustrates the differences between the two types of object detectors. The evaluation metrics for both object detectors are generally evaluated using IoU and mAP. Among them, R-CNN is one of the first object detectors based on deep learning, and ROI is obtained by efficient and simple selective search algorithm. Fast R-CNN is an improvement of R-CNN, which solves the problems of low detection accuracy and slow network reasoning. Fast R-CNN uses convolution during training neural networks to detect the input image and generate ROI projections on feature maps. The ROI is then combined with the feature map for prediction. Fast R-CNN differs from R-CNN in that Fast R-CNN processes the feature map directly with the input image during the detection stage [25]. Faster R-CNN uses a separate detection network, an approach similar to Fast R-CNN, which combines the ROI directly with the ROI pooling layer and feature maps in combination with prediction [26].

Single-stage object detectors are faster than two-stage object detectors because they predict the input object once. YOLOv1 [27] was the first YOLO variant that learned the salient features of objects and could detect them at a faster speed. In 2016, YOLOv2 [28] creates bounding boxes by anchoring them and adds a high-resolution classifier as well as batch normalization. In 2018, Redmon et al. [29] proposed YOLOv3, which consists of a 53-layer backbone network, an independent logical classifier and cross entropy loss to predict bounding boxes and smaller objects. Single-detector SSD model [30] has good inference performance for real-time applications, because it builds object grids in images to generate feature maps. SSD shares features when performing localization and classification tasks on input images. The YOLO model is superior to the SSD in terms of speed but inferior to the SSD in terms of accuracy. Although the SSD and YOLO models have decent inference speeds, there is a class imbalance problem in detecting small objects. To solve this problem, RetinaNet [31] focuses on loss function using a separate network to solve bounding box regression and classification. The performance of each model is summarised in Table 1.

### 2.2. 2D vs. 3D Object Detectors

For object detection, 2D image data is usually obtained by a 2D object detector. In [32], data from radar and camera are fused by learning sensor detection methods. The depth information of objects can play a key role in predicting the position, size and shape of the objects, while 2D object detection can only obtain information in the 2D plane.

Data from radar or laser can be applied to 3D object detectors [33]. These object detectors can use methods such as frustum pointnets [34] and point clouds [35] to predict objects in real-time. In compensating for the loss of object information, some networks often use 2D to 3D augmentation, as it is very expensive and complex to calculate directly using 3D data. The 2D bounding boxes in the dot network are obtained in 2D images and these boxes work well in generating ROIs for 3D object detection, effectively reducing the search effort [36].

With the booming development of deep learning, researchers are increasingly interested in 3D object detection. Complex-YOLO [37] uses the Euler Region Proposal Network (E-RPN) based on YOLOv2 to obtain 3D candidate regions. It achieved 3D object detection and background semantic segmentation by random finite sets (RFS). Wen et al. [38] proposed a lightweight 3D object detection model, which consists of three modules: (1) a point transformation module, which extracts point features from RGB images achieved by the original point cloud; (2) voxelization, which combines the acquired voxel grid with the 3D point cloud to generate a many-to-one mapping; (3) point fusion module, which fuses extracted features for output detection. The performance of these models is summarised in Table 1.

**Table 1 sensors-23-03852-t001:** 2D and 3D object detector models and their performance.

Name	Year	Type	Dataset	mAP	Inference Rate (FPS)
YOLOv1 [27]	2016	2D	Pascal VOC	63.4%	45
YOLOv2 [28]	2016		Pascal VOC	78.6%	67
YOLOv3 [29]	2018		COCO	44.3%	95.2
YOLOv4 [39]	2020		COCO	65.7%	62
YOLOv5 [40]	2021		COCO	56.4%	140
YOLOX [41]	2021		COCO	51.2%	57.8
YOLOR [42]	2021		COCO	74.3%	30
R-CNN [21]	2014		Pascal VOC	66%	0.02
Fast R-CNN [22]	2015		Pascal VOC	68.8%	0.5
Faster R-CNN [23]	2016		COCO	78.9%	7
SSD [30]	2016		Pascal VOC	74.3%	59
RetinaNet [31]	2018		COCO	61.1%	90
Complex-YOLO [37]	2018	3D	KITTI	64%	50.4
Complexer-YOLO [37]	2019		KITTI	49.44%	100
Wen et al. [38]	2021		KITTI	73.76%	17.8

## 3. State-of-Art Methods for MOMCT

Unlike MOT, the camera network consisting of multiple cameras has a much broader view and application prospect than a single camera. This technology specifically contains four key technologies, including object detection, single-camera MOT, vehicle re-identification and multi-camera object tracking association. This section reviews the current state of development in detail and analyses the problems and shortcomings.

### 3.1. Deep Learning Based Object Detection

Object detection is the key part of MOMCT task. The object detector mentioned in Section 2 can be combined with scene classifier, which can learn rich semantic information in images and mine more advanced and deeper features. Frameworks for object detection methods fall into two main types: One is the traditional object detection pipeline, which first generates regional suggestions in the image, and then it classifies each suggestion into different object categories. The other is to treat object detection as a classification or regression problem and then use a unified detection framework to directly derive the final result (location and category). In this section, the object detection task is discussed in terms of the above two types.

#### 3.1.1. Region Proposal Based Framework

The region-proposal based framework has two-step stage, similar to the now prevalent attention mechanism, which first scans the entire image and then focuses on the region of interest. The most prominent of the prior related work [43,44,45] is Overfeat [43]. This model uses CNNs in a sliding window approach to obtain confidence in the underlying object class, then predicts the bounding box from the location of the topmost feature map.

(1)R-CNN

R-CNN achieved an average accuracy of 53.3% on Pascal VOC 2012. It can generate high-quality candidate bounding boxes and then use depth architecture to extract advanced features from images. The main process can be divided into the following three stages.

**Region proposal generation**. The R-CNN combines saliency cues and bottom-up grouping with selective search methods [46]. It is able to generate candidate frames of arbitrary size accurately and quickly, thus reducing the search space in object detection [47,48]. On each image, the R-CNN can provide approximately 2000 region suggestions using the selective search method. Combined with R-CNN, Xie et al. [49] proposed region-oriented proposal networks to generate good region-oriented proposals for identifying them in a cost-free manner. Hong et al. [50] proposed a sparse R-CNN incorporating the Hungarian algorithm to assign learning suggestion frames one-to-one for each positive sample. It generates better features and initial suggestion frames for the training phase.

**CNN based deep feature extraction**. Using the CNN framework, each region is proposed to be tuned to a fixed resolution as the final representation [51]. Due to the remarkable hierarchical structure and expressive power of the neural network, a semantic, robust and high-level feature representation of each region proposal can be obtained. Wang et al. [52] proposed a two-stage detection method for dynamic R-CNNs, using a self-calibrating convolutional module in a convolutional network to extract rich object features. Alsharekh et al. [53] used a deep R-CNN architecture to extract features from the data.

**Classification and localization**. In multiple category pre-training, a multi-category specific linear SVM [54] is used to score different region proposals on a set of negative background regions and positive regions, respectively. Then, within the labelled area, adjustments are made using bounding box regression and filtered using greedy non-maximum suppression (NMS) to produce a final bounding box containing the object location. Zhang et al. [55] proposed a point-to-point regression grouping R-CNN to predict a reasonable bounding box for each point annotation in the image.

(2)R-FCN

Although the Faster R-CNN is an order of magnitude faster than the Fast R-CNN, the computation after the RoI merge layer cannot be shared. Therefore, Dai et al. [56] presented a fully convolutional R-FCN detector to build a shared RoI subnet. However, this simple design has poor detection accuracy, presumably because the deeper network is more sensitive to category semantics. Based on this phenomenon, Vijaya Kumar et al. [57] used the R-FCN ensemble layer to extract predicted scores for each region in the image. This method facilitates the computation of shared regions of interest and improves the detection accuracy of the network. Zhang et al. [58] constructed a region-based full convolution network (R-FCN) model. The model realizes the accurate classification of targets, and improves the recognition accuracy of the model by extracting fine-grained features from images.

#### 3.1.2. Regression/Classification Based Framework

The region-proposal based framework consists of several interrelated phases such as region proposal generation, CNN feature extraction, classification and bounding box regression, which are usually trained separately. Even in R-CNNs with faster end-to-end modules, alternative training is required to obtain the convolution parameters between the detection network and the RPN. In this section, we present two important applications of the framework: YOLO [27] and SSD [30].

(1)YOLO

Redmon et al. [27] proposed YOLO, a method for predicting confidence in bounding boxes and object classes from the topmost feature map. Based on this, Roy et al. [59] proposed a deep learning-based detection model, WilDect-YOLO, in which a residual block was added to the depth space feature extraction. Karaman et al. [60] proposed an artificial bee colony (ABC) optimisation algorithm based on the YOLOv5, which improves the sensitivity of the system for real-time detection by pairing hyperparameters and optimal activation functions. Xue et al. [61] presented an improved YOLOv5, which improves the feature extraction capability by adding self-attentive convolution and convolutional block-attentive modules. Mittal et al. [62] proposed a hybrid model combined YOLO and Fast R-CNN. It employs migration learning methods to reduce class imbalance problems and enhances the image quality through a sharpening process.

(2)SSD

Due to the strong spatial constraints of bounding boxes, YOLO lacks robustness in detecting multiple small objects. It also generates relatively coarse object features during the downsampling operation. To address these issues, Liu et al. [30] proposed SSD based on anchor points employed in RPN [63] and multiscale representation [64]. Contrasted to the fixed grid used in YOLO, SSD can use anchor boxes of different scales to discretize the bounding boxes of a given feature map. The network handles objects with different sizes by fusing the predictions of feature maps in various resolutions. On this basis, Jia et al. [65] used the positive and negative functions of SSD network to solve the shortcoming of insufficient sensitivity for small object detection. Chen et al. [66] enhanced the SSD using MobileNetv2 and the attention mechanism to improve the performance of the algorithm. Gao et al. [67] proposed the R-SSD which based on SSD and ResNet to improve the feature extraction quality of the algorithm. Ma et al. [68] proposed the anchorless 3D object detection model CG-SSD, which mines deeper features through a convolutional backbone network consisting of sparse convolutional layers and residual layers. Cheng et al. [69] used a hybrid attention mechanism in SSD to improve the accuracy of object detection and further combined it with a focal loss function to improve robustness.

### 3.2. Deep Learning Based MOT

MOT needs to be completed after object detection. In this section, we will roughly classify deep learning-based MOT approaches into three categories based on the different tracking framework: (i) MOT using deep network feature enhancement. Deep neural networks are used to extract semantic features for the task of interest and replace the previous traditional manual features. (ii) MOT with deep network embedding. Deep neural networks can classify the different object trajectories acquired and construct deep classifiers to detect whether they belong to the same object. (iii) MOT with end-to-end deep neural network learning. In the general case, there will be intertwined modules in motion object tracking and the MOT results can be obtained directly using deep networks.

#### 3.2.1. MOT Enhancement Using Deep Features

Deep features extracted by deep neural networks are rich in semantic information and are clearly differentiated between different classes. These deep features not only improve the performance of MOT, but are also effective for tasks such as image segmentation and object detection [70].

Similar to object detection using CNN to extract object features [71], AlexNet [72] is mainly used to extract depth features in the multiple hypothesis tracking (MHT) framework. The MHT tracking framework creates a hypothesis tree and uses different associated hypotheses. Kim et al. [73] improved the MHT method with appearance features using regularized least squares. This method reduces the dimensionality of the extracted appearance depth features. Wang et al. [74] proposed a circular tracking unit to model the object by acquiring long-term information. Wojke et al. [75] used the wide residual network (WRN) to extract the depth features in the image to increase the discrimination ability. These features were also used to calculate the minimum cosine distance between tracking and detection. As shown in Figure 5, the whole tracking framework utilizes a cascade matching process. The tracking method has depth features from the WRN that improve the real-time speed of the model while maintaining competitive performance.

The goal of MOT feature learning is to evaluate the similarity between tracking target features, and the CNN structure with two branches is well suited for extracting matching features of moving objects. Leal-Taixé et al. [76] used gradient advancing algorithm to fuse motion and depth features, so that the tracking problem was transformed into linear programming. Zhang et al. [77] proposed a framework that combines filtered tracking and conjoined object tracking. The framework combines artificial features with depth features to improve the performance and robustness of the tracker. Su et al. [78] used anchor-less networks to predict the position of objects in the search element domain, improving the correlation between search frames and template frames.

#### 3.2.2. MOT with Deep Network Embedding

Compared to the enhanced tracking method of depth feature, it is more effective to combine deep neural networks and embedded designs as key components of the tracking framework, where samples of tracking data are used in the training process. Based on the task of network learning, we broadly classify MOT methods for deep network embedding into three types: discriminative deep network learning (DN-MOT), deep measurement learning (DM-MOT), and generative deep network learning (GN-MOT).

DN-MOT: In the DN-MOT method, object trackers optimise discriminative models and search the best locations in the next frames according to the models. Due to deep networks are widely used for discriminative tasks, it is easy to extend discriminative deep network models for tracking tasks. For example, Chen et al. [79] proposed an object particle filtering framework, specifically constructing models of VGG-16 [80] and Faster R-CNN [23] as object classifiers to track each object. Similar to [79], Chu et al. [81] used an object-specific tracker to construct the MOT framework and used an updated classifier to find the best candidate to complete the tracking, as shown in Figure 6. Firstly, the captured video frames are fed into a shared CNN layer to generate feature maps, which are subsequently fed into a RoI pool to generate candidate features. At the same time, positive, negative and historical samples are selected from the tracked samples. These samples are used to calculate the classification score and weight loss of candidate features. Secondly, the features in the RoI pool are passed through a fully-connected convolutional layer to generate a visibility map, and then spatial attention is extracted from the visibility map. Finally, the state of candidate features is extracted for object tracking. In addition, Liu et al. [82] provided a multivariate polynomial kinematic forward solution algorithm that effectively improves the accuracy and real-time performance of the model.

DM-MOT: It involves extracting image features and mapping them to a high-dimensional space. This method aims to calculate the similarity between different objects based on the feature information. It can be regarded as an image block verification problem [83]. Similar to pedestrian re-identification [84] and face recognition [85], it is important to learn accurate distance metrics and similarity models in such problems. Therefore, it is desirable to use suitable depth metric learning networks [86] for DM-MOT. Son et al. [87] used multiple image blocks as Siamese network inputs to achieve precise localisation by extracting appearance and motion features. Xiang et al. [88] designed a CNN network based on triadic loss to acquire the appearance features of the object and through, which the distance metric between the detector and the tracker was learned and calculated. Unlike [87], Cheng et al. [89] employed the appearance and motion features of the object as input for DM-MOT and used triple loss to optimize the model. In general, it usually applies the Hungarian algorithm [90] to solve the distance metric cost incurred by the detectors and trackers during the tracking process.

GN-MOT: It aims to generate the object model by learning its shape, motion and other characteristics and then using this model to track its trajectory. Some works have used deep generative learning to improve the performance of MOT. Fang et al. [91] presented a recursive auto-regressive network (RAN) model to improve the performance by modeling the motion features. Fernando et al. [92] predicted object trajectories by using an LSTM-based generative model. It associates the object tracker with the GAN model in the prediction module to enable the tracking of new objects.

#### 3.2.3. MOT with End-to-End Deep Network Learning

It is difficult to apply a single model for learning multiple key modules in an MOT task such as target detection, target matching, trajectory tracking etc. Recently, researchers have presented many end-to-end learning methods to achieve this goal. Inspired by Posner and Ondruska [93], Milan et al. [94] modeled these procedures on the basis of RNNs. As shown in Figure 7, the inputs to the network are the matching matrix, state, and presence probability of the object, while the updated result, predicted state, and new presence probability are used as outputs, which determine whether the object should be terminated. In addition, an LSTM-based network was designed for computing the matching matrix, while modeling the matching process between the current observation and the object state to train the RNN in an end-to-end manner. On this basis, Sadeghian et al. [95] employed a hierarchical RNN structured network to obtain interaction, appearance and motion features of objects. The network uses an end-to-end approach to train matching classifiers, which improves the probability of matching between objects and trackers. With a time-synchronous stability mechanism, Li et al. [96] enabled all parts of the controller’s agent states to reach consensus simultaneously, effectively improving the performance of the model. Moreover, Kim et al. [97] proposed an end-to-end training method to improve the performance by using bilinear LSTM module.

### 3.3. Deep -Learning-Based Vehicle Re-Identification

Following single camera multi-objective tracking, a deep learning based re-identification method is used to extract embedded features of each trajectory for vehicle re-identification. This section will focus on unsupervised learning-based methods, attention mechanism-based methods and local feature-based methods.

#### 3.3.1. Vehicle Re-Identification Based on Unsupervised Learning

Compared to supervised techniques, unsupervised learning aims to make inferences directly from unlabeled input data, which addresses the limited generalisation capability of the model and the high cost of manually labelled data [98]. This technology has been widely used in vehicle re-ID task. Deng et al. [99] presented a cross-domain adaptive unsupervised method from image to image. This method maintains similarity by combining neural network and improves the recognition accuracy of the model. Wang et al. [100] used an identity-based joint attribute learning method to improves the efficiency by the key attributes and semantic information.

In recent years, unsupervised methods have been widely used in vehicle re-identification. Shen et al. [101] used clustering features to simulate global and local features for improving the accuracy of unsupervised vehicle re-ID. Zhu et al. [102] trained convolutional neural networks with a deep feature learning module to improve the model’s ability in feature differentiation. Wang et al. [103] proposed a new contrast learning framework that uses reliable discrete sample clustering to build a memory dictionary for object re-identification. Gao et al. [104] used an unsupervised framework based on data synthesis. By adjusting the target and source domain to adapt to the pre-training model, the domain generalization ability of the re-recognition model was improved.

Generated Antagonistic Network (GAN) [105] is widely used in unsupervised learning. A simulation of the GAN framework is shown in Figure 8. The framework consists of a data discriminator and a generator. The generator obtains the synthetic data through transformation after obtaining the random variables. The discriminator receives the data from the generator and judges the data, and finally reaches a balanced state. Zhou et al. [106] presented a GAN-siame network to solve the unsupervised V-reID cross-domain problem. The algorithm learns the distance measurement between two domains by connecting the network, which improves the performance of model matching.

#### 3.3.2. Vehicle Re-Identification Based on Attention Mechanism

Attention mechanism mainly focuses on selective actions/things related to tasks and ignores other irrelevant actions/things. Researchers are working on designing an effective attention-based neural network for vision-related applications such as fine-grained image recognition [107,108], image classification [109,110], image captioning [111,112], and vehicle re-identification [113]. The process of vehicle re-identification based on spatio-temporal attention is shown in Figure 9. Trajectory features are extracted for re-ID by the spatial attention mechanism, then the features are weight ranked by the temporal attention mechanism, and finally the key features of objects are output according to the ranking.

These approaches typically follow a strategy of integrating hard partial selection subnets or soft mask branches into deep networks. For example, Zhu et al. [113] added self-attentive models to each branch of the CNN network for fine-grained recognition of vehicles. To reduce the influence of noise in the image, Lian et al. [114] used the attention network based on transformer to extract the global features of vehicle re-ID. Jiang et al. [115] studied a global reference attention network. By mining distinguishing features, the difficulty of distinguishing vehicles with similar appearance is solved. Tian et al. [116] proposed an adaptive attention network. The network captures the global structural information of the vehicle through a global relational attention module to improve the accuracy of re-ID. Li et al. [117] investigated a CAM network with a contrast attention module. It enhanced the recognition ability of the re-ID model by refining the local features. Song et al. [118] introduced the global attention mechanism based on two-branch network. It trains the network by combining global and local features to improve the performance of vehicle recognition. Li et al. [119] presented a model with region attention and orthogonal view generation. The process of re-identification is simplified by extracting distinguishable regional features to differentiate vehicles. Tang et al. [120] studied an attention network for extracting multi-scale features, which reduced the difficulty of vehicle re-ID task. Liu et al. [121] used a multiple soft attention network to extract robust features. Shen et al. [122] presented a graphical interaction converter to extract differentiated local features for the robustness of re-ID model.

#### 3.3.3. Vehicle Re-Identification Based on Local Feature

Early research on vehicle re-ID focused on the global features of images. After encountering the bottleneck of accuracy, many studies began to pay attention to local features, because the differences between similar vehicles are mainly reflected in local areas. At present, it is common to extract local features by means of key point localisation and region segmentation. Wang et al. [123] segmented the image vertically and horizontally to extract features, which improves the accuracy of vehicle re-recognition. Rong et al. [124] fuse local-global features to obtain more vehicle information and enhance the learning ability of vehicle recognition. Yang et al. [125] studied the two-branch network based on pyramid feature learning. It solves the problem of learning and recognising model information by learning local and global features of the vehicle. Fu et al. [126] utilized local attention to facilitate the learning of local attentional features for vehicle re-ID. Liu et al. [127] designed a vehicle information module. The module improves the ability to recognize similar vehicles under the same camera.

### 3.4. Deep Learning Based Multi-Object Multi-Camera Tracking

Re-ID is carried out through the information of cameras, and the cross-regional tracking of moving objects is completed through the temporal and spatial correlation of multi-camera trajectories. The MOMCT with trajectory to object method is as shown in Figure 10. Input the object information detected in different cameras into the camera network, and then accurately match the object trajectories, and create a complete global cross-camera trajectory for each object.

Vehicle trajectory matching is the key part of MOMCT. Due to variations in lighting and viewing angles, it is susceptible to disruptions in vehicle tracking trajectories while vehicles are obscured and hence continuous vehicle tracking cannot be accomplished. For such problems, Hsu et al. [128] proposed a camera linkage model based on trajectory. By extracting the appearance and topological features of different cameras, the accuracy of vehicle trajectory matching is improved. Hsu et al. [129] provided a reliable framework for vehicle MOMCT using a hierarchical clustering algorithm. Li et al. [130] simplified the process of object trajectory matching in overlapping space by dynamically coding visual features. Liu et al. [131] used Markov decision to model the vehicle trajectory, which improves the accuracy of trajectory matching. Zhao et al. [132] presented a channel estimation method which uses sensing, communication and control technologies to obtain the information needed for trajectory generation.

During tracking, it is easy to mistake tracks generated by different vehicles as the same ID. To solve this problem, Yang et al. [133] designed the trajectory re-connection technology. By reconnecting the segmented trajectories, an accurate vehicle trajectory is generated. Li et al. [134] developed a MOMCT vehicle tracking system which eliminates unreliable trajectories by assessing cross-view matching of vehicle trajectories. Liang et al. [135] used Kalman filter to predict the motion of the object, which improved the analysis and matching ability of the model. To match the local trajectory of the same object in different cameras, He et al. [136] employed the spatio-temporal attention mechanism to generate the vehicle trajectory representation, which improves the matching success rate of the object allocation algorithm. Tran et al. [137] presented a spatially constrained framework to improve the robustness of the model by using cross-awareness of the tracker.

## 4. Datasets

Datasets play an important role in the MOMCT task, not only to strategise and compare the performance of various algorithms, but also to help solve complex and challenging problems in the field. This section describes the main datasets in the MOMCT task.

### 4.1. Analysis of Public MOMCT Datasets

#### 4.1.1. BDD100K Dataset

The BDD100K [138] dataset contains 100,000 videos with IMU/GPS information captured by mobile phones, each video lasts approximately 40 s at 30 frames per second and keyframes are extracted and annotated at the 10th second, mainly in terms of road object boundaries, driveable areas, and lane markings. The dataset annotates the boundary boxes for common objects on the road in 100,000 keyframe images to understand the location and distribution of the objects. The different traffic scenes included in the BDD100K dataset are shown in Figure 11.

#### 4.1.2. VehicleX Dataset

The VehicleX dataset [139] is a dataset synthesized from 3D models of various vehicles. It is taken from real-world scenes and used to synthesize images with a total of 1362 vehicles and 192,150 images. In addition, colour and type labels were also annotated. The traffic scene images of vehiclex dataset is shown in Figure 12.

#### 4.1.3. UA-DETRAC Dataset

The UA-DETRAC data set [140] contains 10 h of video from 24 different locations, including more than 140,000 frames of data. Up to 8250 vehicle objects in the scene are manually marked with more than 1.21 million object bounding boxes containing labels. The vehicle categories in the data set are cars, buses and trucks, and there are also four weather types: cloudy, night, sunny and rainy. As shown in Figure 13, UA-DETRAC dataset images with different congestion situations in traffic scenes.

#### 4.1.4. KITTI Dataset

The KITTI dataset [141] was used to evaluate the performance of computer vision techniques such as stereo imagery, optical flow, visual ranging, 3D object detection, and 3D tracking in an in-vehicle environment. It contains real image data collected from urban, and rural and motorway scenes with up to 15 vehicles and 30 pedestrians per image, as well as various levels of occlusion and truncation. The traffic scene images of KITTI dataset is shown in Figure 14.

#### 4.1.5. Nuscenes Dataset

The nuScenes dataset [142] is a shared large dataset for autonomous driving. The dataset has 1000 driving scenarios, each with 20 s of video, for a total of approximately 15 h of driving data. The scenarios were selected with due consideration for diverse driving maneuvers, traffic conditions, and accidents. The traffic scene images of nuscenes dataset is shown in Figure 15.

### 4.2. Summary of Typical Datasets

With the detailed description of the data related to the MOMCT dataset above, this paper summarises and tabulates the vehicle dataset in recent years, as shown in Table 2.

## 5. Evaluation Metrics

Certain evaluation metrics can measure the performance of cross-camera MOT tasks. It plays a key role in the evaluation analysis and selection of algorithms as a criterion for evaluating the performance of MOMCT algorithms. In this section, the MOMCT algorithm performance evaluation metrics are described in detail.

### 5.1. Basic Evaluation Metrics

(1)TP: True Positive is a positive sample that is predicted to be positive by the model, which can be referred to as the percentage of correct judgments that are positive.(2)TN: True Negative is a negative sample that is predicted to be negative by the model and can be referred to as the percentage of correct judgments that are negative.(3)FP: False Positive is a negative sample that is predicted to be positive by the model and can be referred to as the false positive rate.(4)FN: False Negative refers to positive samples that are predicted to be negative by the model and can be referred to as the under-reporting rate.(5)Accuracy: This refers to the weighting of the correct decision by the classifier, and is publicly expressed as.
(1)$$A=\frac{TP+TN}{TP+TN+FP+FN}.$$(6)Precision: Its the proportion of true positive samples among the positive examples determined by the classifier, expressed publicly as.
(2)$$P=\frac{TP}{TP+FN}.$$(7)Recall: Its the proportion of positive cases correctly determined by the classifier to the total number of positive cases, expressed publicly as.
(3)$$R=\frac{TP}{TP+FN}.$$

### 5.2. Track Relevant Indicators

(1)MOTA [143]: MOT Accuracy is a measure of single-camera MOT accuracy and is publicly represented as.
(4)$$\mathrm{MOTA}=1-\frac{FN+FP+\Phi }{T},$$
where FN is the sum of the false negatives of all frames, i.e., assuming that $fn_{t}$ is the false negative of frame *t*, then $FN=\sum_{t}fn_{t}$. Similarly, $FP=\sum_{t}fp_{t}$. *T* is the sum of the number of real objects in all frames, i.e., assuming that there are $g_{t}$ objects in frame *t*, then $T=\sum_{t}g_{t}$. Φ is the number of object jumps in all frames, $\phi_{t}$ is the number of object jumps in frame *t*, then $\Phi =\sum_{t}\phi_{t}$. In other words, these three items represent the missing rate, the false positive rate, and the mismatch rate in that order. The closer MOTA is to 1 the better the tracker performance.(2)MOTP [143]: MOT accuracy is a measure of single-camera MOT position error, expressed by the formula.
(5)$$\mathrm{MOTP}=\frac{\sum_{i,t}d_{t}^{i}}{\sum_{t}c_{t}},$$
where $c_{t}$ denotes the number of matches at frame *t*. For each pair of matches, the matching error $d_{t}^{i}$, represents the distance between the object $O_{i}$, and its pairing hypothesis position at frame *t*.(3)MT: Mostly tracked is the number of tracks where the tracked portion is greater than 80%, the larger the value the better.(4)ML: Mostly lost is the number of tracks where the lost portion is greater than 80%, the smaller the value the better.(5)Frag: The number of jumps is the number of track changes from “tracking” to “not tracking” state.

### 5.3. ID Related Index

(1)IDP: Identification Precision is the accuracy of vehicle ID identification in each bounding box. The formula is:
(6)$$\mathrm{IDP}=\frac{IDTP}{IDTP+IDFP},$$
where IDIP and IDFP are the number of true IDs and the number of false positive IDs, respectively.(2)IDR: Identification Recall is the recall rate of vehicle ID identification in each bounding box. The formula is:
(7)$$\mathrm{IDR}=\frac{IDTP}{IDTP+IDFN},$$
where IDFN is the negative ID number.(3)IDF1: Identification F-Score is the F-value of the vehicle ID identification in each bounding box. The formula is:
(8)$${\mathrm{IDF}}_{1}=\frac{2IDTP}{2IDTP+IDFP+IDFN}.$$In general, IDF1 is the first default metric used to evaluate the performance of the tracker. These three metrics can be inferred from any two of them, so it is also possible to show only two of them, although it is preferable that these two include IDF1.(4)IDS: The number of ID switches is the number of instantaneous vehicle ID transitions in the tracking track, usually reflecting the stability of the tracking, the smaller the value the better.

## 6. Typical Algorithms and Visualization Results

### 6.1. Comparison and Analysis of Algorithms

To describe MOMCT algorithm models more intuitively in Section 3, these algorithm models are listed in Table 3. It not only shows their performance of using different object detectors and tracking methods on BDD100K data set, but also shows their results based on IDP, IDR, IDF1 and other indicators on the data set, and makes the following analysis and comparison of typical algorithms in Section 3.

By comparing these algorithms, we can draw the following summary:(1)GCNM: After associating the object trajectory, the algorithm uses the graph convolution network to form the global trajectory. Then on the trajectory level, instant erasure and random horizontal flip are used to expand the data, which enhances the data robustness of the camera. Finally, the new loss function improves the generalization ability of the model, thus obtaining good performance in data accuracy.(2)UWIPL: This method generates a motion track by using its appearance and time information. The system takes ResNet50 as the backbone network and combines Xent loss and Htri loss for training. The tracking accuracy is improved based on road channelization and road condition information, and the applicability of this method in different scenarios is realized.(3)ANU: It provides fine-grained features by using road spatio-temporal information and camera topology information. Removing overlapping bounding boxes by non-maximum suppression. At the same time, it also uses the color dithering mechanism to improve the performance of the model.(4)BUPT: It utilizes ResNet network as the backbone network and uses random filling and erasing methods to fill data. Then, it trains the framework by combining trajectory consistency loss and clustering loss. Finally, a higher IDF1 is obtained by introducing temporal and spatial clues.(5)DyGLIP: It has better lost trajectory recovery and better feature representation during camera overload. By adding correlation regression and attention module in the experiment, the scalability of the model in large-scale data sets is improved.(6)Online-MTMC: It solves the MOMCT problem by using the detection-clustering method. The feature pyramid network is used as the backbone network, and the quality of features is improved by Gaussian blur and contrast disturbance mechanism. This method also employs the minimum loss function to optimize the network parameters.(7)ELECTRICITY: It applies a cluster loss strategy to remove isolated tracks and synchronise track ID based on the re-identification results. Meanwhile, depth ranking is considered as a tracking model and Adagrad is applied as a loss function to optimise the model, which makes the algorithm suitable for large-scale realistic intelligent traffic scenes.(8)NCCU: It adopts vehicle image features and geometric factors for collaborative optimization matching. Then, FBG analysis is used to generate the mask of road region of interest, which effectively solves the problem of finding broken down vehicles on the road.

### 6.2. Visualization Results and Analysis

The visualization results of the listed classical algorithms are shown from Figure 16, Figure 17, Figure 18 and Figure 19. Through visual analysis, we can get the following results.

(1)ANU adjusts the thresholds of positive and negative sample pairs by increasing the perception of locality in small scenes. The non-maximum suppression mechanism also removes some of the overlapping bounding boxes and retains those close to the camera, improving the success rate of tracking to the vehicle.(2)UWIPL combines camera linking and deep feature re-identification of trajectories, uses the appearance and time information of trajectories for high confidence matching, and uses a greedy algorithm to select the smallest pairwise distance to match the vehicle being tracked, resulting in accurate tracking results in different scenarios.(3)ELECTRICITY combines MOMCT strategy and aggregation loss to eliminate the erroneous trajectories. It tracks objects mainly through re-identification, and further improves the robustness of the algorithm through image flipping and random erasure.(4)BUPT system combines the loss of trajectory consistency with the loss of clustering, and extracts more obvious features. The cluster loss used in the method improves the tracking accuracy.

## 7. Challenges, Applications and Perspectives

### 7.1. Challenges and Opportunities

(1)Real-time processing

In the training stage, object detection in the sequences increases the training time. It can better detect new objects by increasing the correlation in the sequences, while discarding irrelevant frames. It may reduce the waiting time in training. It has become an open question that how to establish spatio-temporal relationships between consecutive frames in order to improve the efficiency of detecting objects. Although some recent research works have begun to address this problem, there is still a great demand for further works.

(2)Semi-supervised object detection

The existing methods in object detection basically need to label data sets to train models. The supervised object detection model faces the challenge of the change of labeled data caused by scene change. Due to the changing nature of object tracking and detection, introducing semi-supervised learning into the model can effectively reduce the training time of object detection. Researchers recently used semi-supervised transformer model to improve the accuracy of object detection. However, it is still a challenge to apply them to the detection model, because they occupy a lot of memory and need further study.

(3)Publicly available datasets

Changing environmental conditions (such as weather) affect the performance of the object detector, but data from different environments can improve this situation during training. Using new data to shape and test the model can make the model adapt to the changes of weather environment. Therefore, there is a fundamental requirement for a dataset containing a wide range of data to train the model in order to ensure better robustness.

### 7.2. Applications

With the advancement and development of deep learning and various new technologies in recent years, the application of MOMCT in smart city has become more widespread and encompasses all aspects of life.

(1)Intelligent transportation

MOMCT technology enables real-time continuous tracking of vehicles on highways and is able to solve practical tracking problems in complex environments, such as occlusion, weather, light changes, etc. This technology obtains the traffic parameters of the vehicle and enables high precision continuous tracking of the vehicle on the highway. MOMCT technology can get the data of road congestion in time and provide people with more convenient travel modes. Meanwhile, traffic workers can monitor key road conditions by using MOMCT technology. In case of accidents such as traffic accidents, report to the traffic police in time and inform other car owners who are about to pass the accident section to avoid. It not only improves the efficiency of accident handling, but also reduces the waiting time for other travelers.

(2)Intelligent surveillance

Traditional monitoring technology usually consumes a lot of human and material resources, which is not in line with the development direction of the times. Because of its flexibility and accuracy, MOMCT technology is widely used for real-time monitoring in scenic spots, hospitals, banks, supermarkets and other public places with high traffic. This technology can track and detect moving objects such as pedestrians and vehicles in real time, and it identify objects according to semantic information such as facial expressions and postures.

(3)Automated driving

With the development of technology, advanced autonomous driving has received extensive attention. This technology brings convenience and reduces the probability of accidents. The application of MOMCT technology in autonomous driving makes self-driving cars popular. It uses radar sensors and laser rangefinders to observe the surrounding traffic conditions, and keeps real-time, multi-directional and multi-viewpoint attention to moving objects and changes in the surrounding environment. It is realized that the vehicle keeps a relatively safe distance from other objects at all times, and traffic accidents can be effectively avoided.

### 7.3. Outlook

(1)Learning-based active tracking

Currently, some methods deal with some occlusion and ambiguity by reducing the reliance on trackers. However, they still need each camera to track the object, and then make motion control judgment and calculation. This method is easily influenced by the tracker. At present, it is difficult to obtain real value through labelling, so it is worth exploring further how to construct effective incentive functions in the real world.

(2)Multi-view information fusion

Some methods combine multi-view single object tracking methods to fuse information by directly combining features. Although these methods learn a more complete object image via a multi-view object model to achieve object tracking, camera movements can lead to image blurring. At the moment, there is no well-developed solution to solve this problem and it still requires continuous exploration by researchers.

## 8. Conclusions

In this paper, we present a review of recent advances in techniques and algorithms related to deep learning for multi-object multi-camera tracking tasks, including object trackers for MOMCT, analysis of different types of MOMCT methods, benchmark datasets, and evaluation metrics. In addition, several classical approaches and visualization results are presented to compare their performance. Although research on MOMCT tasks has made great progress in recent years, there are still significant challenges such as real-time processing problems, semi-supervised object detection problems, and open dataset problems. Therefore, we provide some perspectives on the future direction of MOMCT, including learning-based active tracking, multi-view information fusion, 3D object tracking, etc. The practical application of MOMCT in smart cities can both contribute to technological reform and help people create a better life. It is believed that this paper can help researchers gain insight into MOMCT and its application in practical scenarios, thus furthering its progress and development.

## Figures and Tables

**Figure 1 sensors-23-03852-f001:**
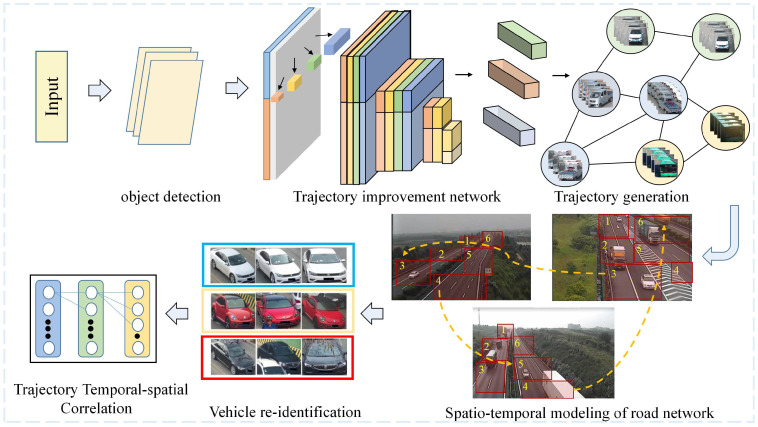
Cross-area tracking of vehicles with multiple objects and cameras.

**Figure 3 sensors-23-03852-f003:**
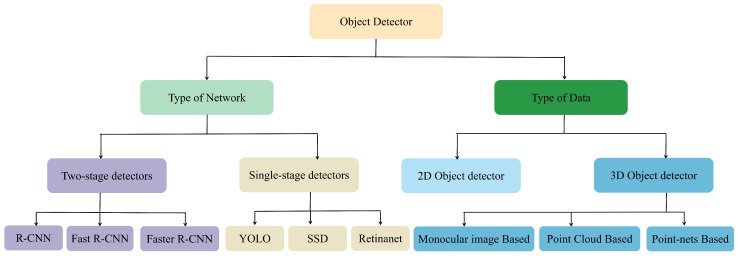
Classification of object detectors.

**Figure 4 sensors-23-03852-f004:**
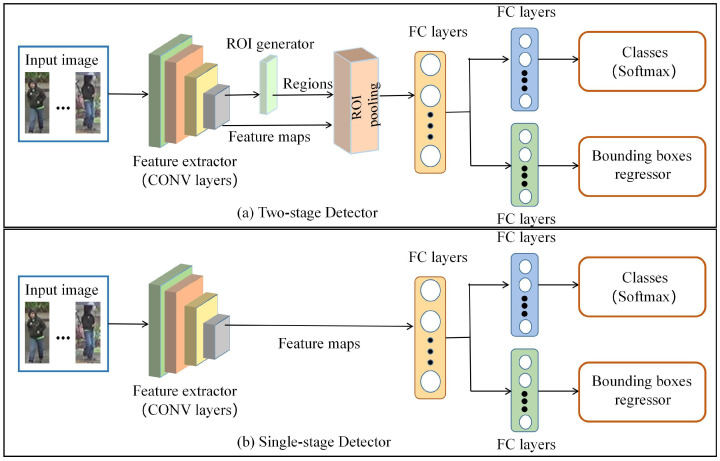
Two-stage vs. Single-stage object detector diagram.

**Figure 5 sensors-23-03852-f005:**
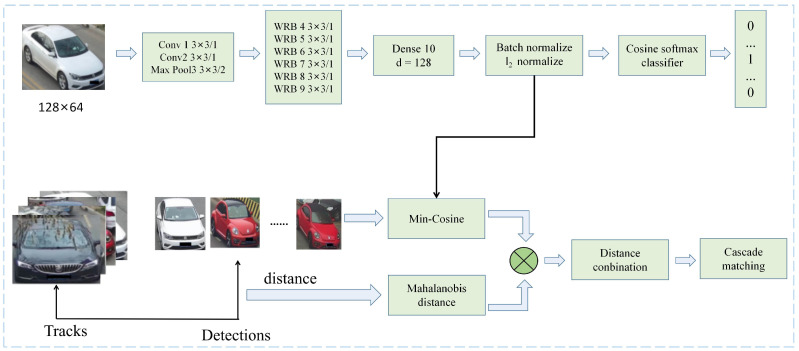
The framework of depth SORT [75].

**Figure 6 sensors-23-03852-f006:**
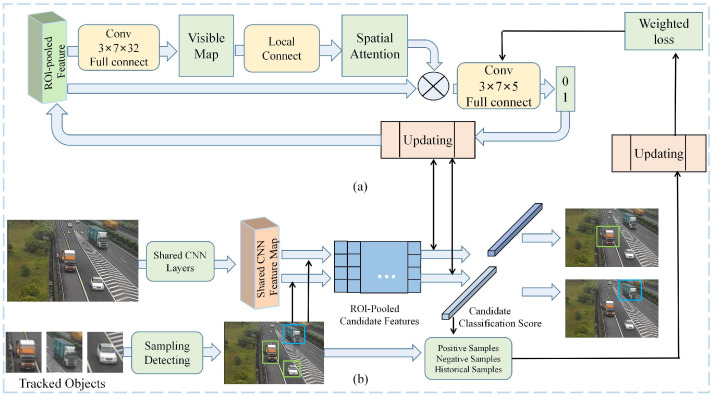
Framework of STAM-MOT [81]. In this framework, (**a**) a deep neural network-based spatial attention and object-specific classifier, and (**b**) a sampling-based best candidate classifier.

**Figure 7 sensors-23-03852-f007:**
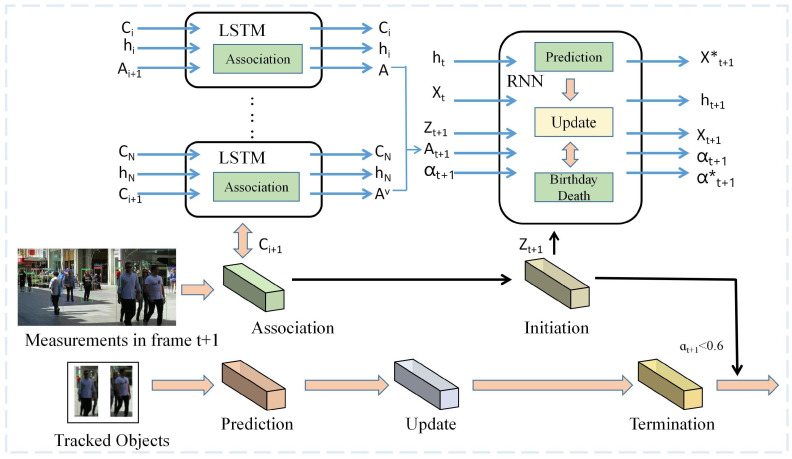
Framework of RNN-LSTM tracking [94]. An LSTM-based network is constructed in this framework for deriving the best association between object and detection, and an RNN-based network for updating states, learning predictions and termination probabilities.

**Figure 8 sensors-23-03852-f008:**
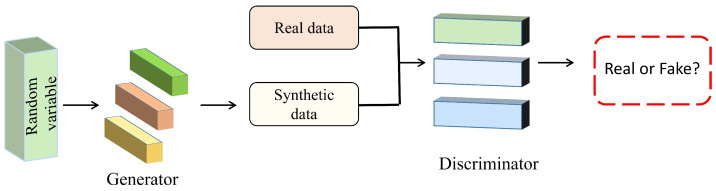
GAN architecture simulation diagram.

**Figure 9 sensors-23-03852-f009:**
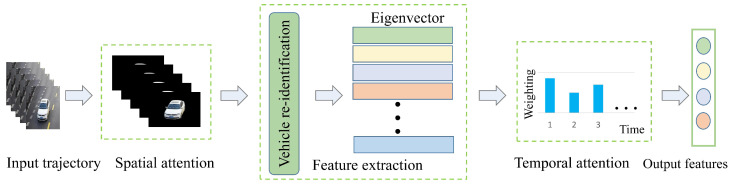
Vehicle Re-identification Based on Spatio-temporal Attention.

**Figure 10 sensors-23-03852-f010:**
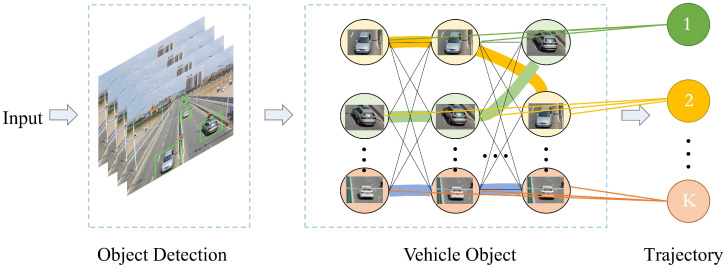
Schematic diagram of algorithm from trajectory to object.

**Figure 11 sensors-23-03852-f011:**
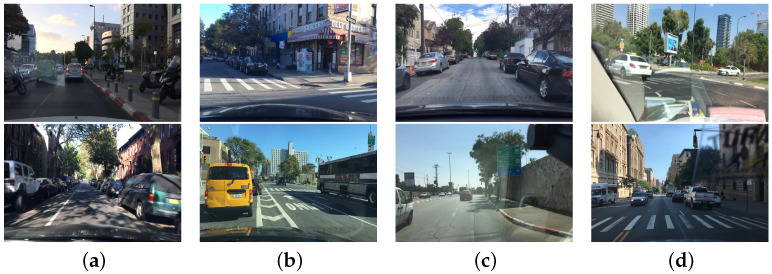
Traffic scene images of BDD100K data set, where (**a**–**d**) are images of traffic captured in different street scenes.

**Figure 12 sensors-23-03852-f012:**
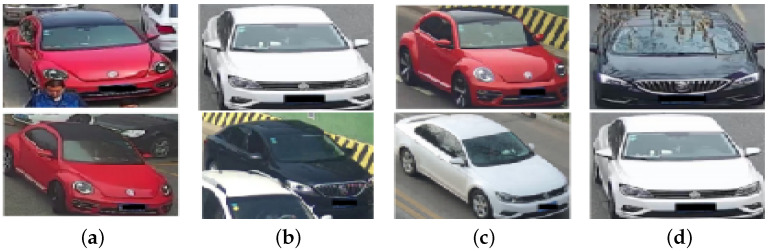
Traffic scene images of UA-DETRAC data set, where (**a**–**d**) are images of different vehicles in a traffic scene.

**Figure 13 sensors-23-03852-f013:**
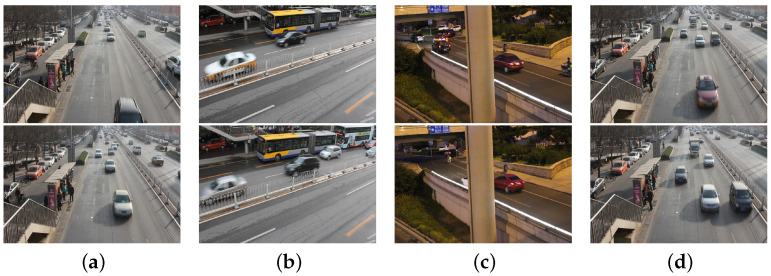
Traffic scene images of UA-DETRAC data set. (**a**,**b**,**d**) show the traffic scenes in daytime conditions. (**c**) shows the traffic scenes on the viaduct under night conditions.

**Figure 14 sensors-23-03852-f014:**
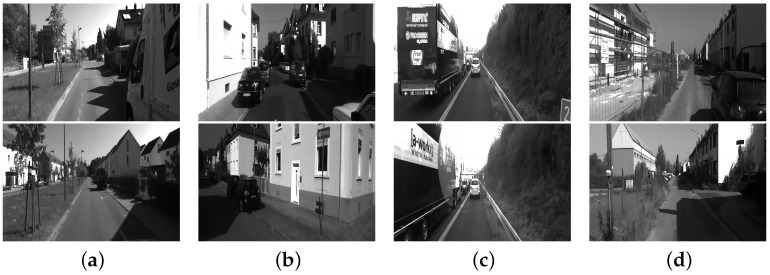
Traffic scene images of KITTI data set, where (**a**–**d**) are images of different urban and rural roads.

**Figure 15 sensors-23-03852-f015:**
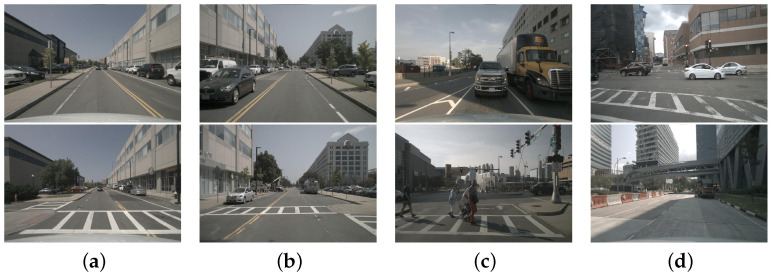
Traffic scene images of nuscenes data set, where (**a**–**d**) are images of intersections, office buildings and residences on different streets.

**Figure 16 sensors-23-03852-f016:**
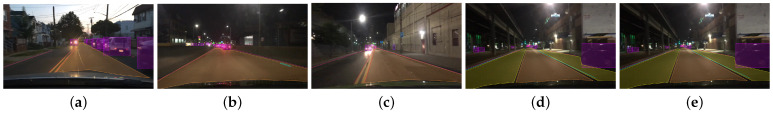
Visualizatio n results of ANU tracking system on BDD100K data set. (**a**) shows the results of vehicle detection and tracking during the day, (**b**–**e**) show the results of vehicle detection and tracking at different times of the night for the same video.

**Figure 17 sensors-23-03852-f017:**
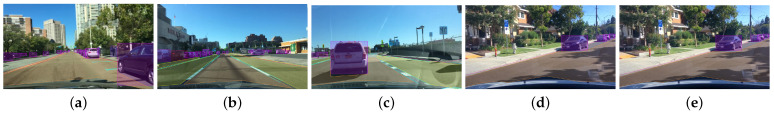
Visualization results of UWIPT tracking network on BDD100K data set, where (**a**–**e**) show the results of detecting and tracking vehicles in different scenes of the same video.

**Figure 18 sensors-23-03852-f018:**
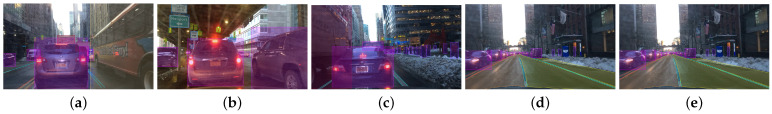
Visualization results of ELECTRICITY tracking algorithm on BDD100K data set, where (**a**–**e**) are the results of vehicle detection and tracking in the same scene in the same video.

**Figure 19 sensors-23-03852-f019:**
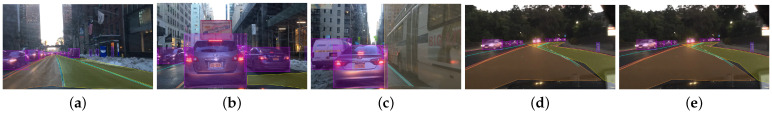
Visualization results of BUPT system on BDD100K data set. (**a**–**c**) are urban scenes, (**d**,**e**) are countryside scenes.

**Table 2 sensors-23-03852-t002:** Popular dataset for MOMCT.

Dataset	Year	Total Images	Categories	Image Size	Objects of Image	Size	Highlights
OpenData	2017	10,000	10	400 × 424	Varied	16 G	A great variety.
Stanford Cars	2013	16,185	5	720 × 540	3	10 G	Automobile model verification.
CompCars	2015	136,726	5	540 × 540	4	18 G	Fine-grained classification.
ImageNet	2009	14,197,122	21,841	500 × 400	1.5	138 G	Image classification, detection and location.
PASCAL VOC	2009	11,540	20	470 × 380	2.4	8 G	One of the mainstream data sets of computer vision.
MS COCO	2015	328,000+	91	640 × 480	7.3	18 G	Very high industry status and huge data set.
Open image	2020	9 million+	6000+	Varied	8.3	500 G	Very diverse.
KITTI	2012	500+	5	1240 × 376	1.7	180 G	Evaluate vehicle performance.
BD100K	2018	100,000	10	1280 × 720	2.4	7 G	One of the largest driving data sets.
UA-DETRAC	2020	140,000	8	960 × 540	2.3	14.5 G	Challenging data set.
ILSVRC	2012	170,000+	1000	1280 × 720	Varied	16 G	The most popular machine vision competition.
vehiclex	2020	192,150	10	960 × 540	Varied	16 G	Accurate data.
CityFlow	2019	229,680	6	1080 × 540	Varied	8 G	Large-scale.
VehicleID	2019	221,763	11	840 × 840	Varied	7 G	Large-scale.

**Table 3 sensors-23-03852-t003:** Performance of Typical Algorithms on BDD100K Datasets.

Method	Object Detector	SCT	IDP↑	IDR↑	IDF1↑
GCNM [144]	SSD	TNT	71.95	92.81	81.06
UWIPL [145]	SSD	TNT	70.21	92.61	79.87
ANU [146]	SSD	custom	67.53	81.99	74.06
BUPT [147]	FPN	custom	78.23	63.69	70.22
DyGLIP [148]	Mask-RCCN	DeepSORT	-	-	64.90
Online-MTMC [149]	EfficientDet	Custom	55.15	76.98	64.26
ELECTRICITY [150]	Mask-RCCN	DeepSORT	-	-	53.80
NCCU [151]	FPN	DaSiamRPN	48.91	43.35	45.97

IDP: the accuracy of vehicle ID identification in each bounding box. IDR: the recall rate of vehicle ID identification. IDF1: the F-value of the vehicle ID identification in each bounding box.

## Data Availability

The datasets are available on Github at https://github.com/wwwj795/datasets, accessed on 14 March 2023.

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
