# Peer review of "Multi-Object Multi-Camera Tracking Based on Deep Learning for Intelligent Transportation: A Review"

_sensors, 2023, doi:10.3390/s23083852_

Round 1
Reviewer 1 Report
In the article, the authors described the Multi-Target Multi-Camera Tracking technology based on Deep learning in intelligent transportation. The article is an overview, presenting a wide range of issues, which is well illustrated in Figure 2. Despite the comprehensive scope of the topic, the authors managed to sketch some issues a little closer, such as Deep Learning Based Multi-object Tracking. The authors also pointed to publicly available large databases, which adds to the value of the publication. It can be concluded that the review was done carefully and with a lot of effort, which is confirmed by a very extensive list of bibliographies.
Unfortunately, there are quite a few minor editing mistakes in the article. Hence it needs to be reread carefully and corrections made. Here are some examples:
- line 32/33 - aspects are not components (it should say "components")
- Fig. 2 - "Multi-object racking..." -> should be "tracking"
- Fig. 3 - lower row ob blocks - the font used is too small
- line 317 - "leal" is a beginning of a new sentence - it should start with a capital letter
- lines 349-355 -> a sentence is repeated twice
lines 356-358 -> the sentence is not clead -> it should be rewritten
lines 366-367 - the sentence should be rewritten
line 407 - it should say "Compared to..."
line 432 - start a new sentence with a capital letter
line 476 - space is needed after "[120"
line 513 - "li" is a name - it should be "Li"
line 637 - It seems that points (1) to (8) contain a rather abbreviated characterization and do not have the character of a conclusion. Probably "summary" is more appropriate.
line 678 - similarly as above
...and other similar issues should be corrected until the paper is ready for publishing.
Reviewer 2 Report
1. motivation of the review is not clear
2. authors should clearly state how your work is better than other related reviews in this area?
3. how the authors did perform comparison with similar works?
4. some latest works needs to be included
1. Extendable Multiple Nodes Recurrent Tracking Framework with RTU++. IEEE Transactions on Image Processing. Doi: 10.1109/TIP.2022.3192706
2. On Model Identification Based Optimal Control and It’s Applications to Multi-Agent Learning and Control. Mathematics, 11(4). doi: 10.3390/math11040906
3. Integrated Sensing and Communications for UAV Communications with Jittering Effect. IEEE Wireless Communications Letters. doi: 10.1109/LWC.2023.3243590
4. Fixed-Time-Synchronized Consensus Control of Multiagent Systems. IEEE transactions on control of network systems, 8(1), 89-98. doi: 10.1109/TCNS.2020.3034523
5. Kinematics Model Optimization Algorithm for Six Degrees of Freedom Parallel Platform. Applied Sciences, 13(5). doi: 10.3390/app13053082
